# Ice front retreat reconfigures meltwater-driven gyres modulating ocean heat delivery to an Antarctic ice shelf

Seung-Tae Yoon[1], Won Sang Lee [2✉], SungHyun Nam [3✉], Choon-Ki Lee[2], Sukyoung Yun[2], Karen Heywood [4], Lars Boehme [5], Yixi Zheng [4], Inhee Lee[6], Yeon Choi[3], Adrian Jenkins [7], Emilia Kyung Jin[2], Robert Larter [8], Julia Wellner[9], Pierre Dutrieux [8] & Alexander T. Bradley [8]

Pine Island Ice Shelf (PIIS) buttresses the Pine Island Glacier, the key contributor to sea-level rise. PIIS has thinned owing to ocean-driven melting, and its calving front has retreated, leading to buttressing loss. PIIS melting depends primarily on the thermocline variability in its front. Furthermore, local ocean circulation shifts adjust heat transport within Pine Island Bay (PIB), yet oceanic processes underlying the ice front retreat remain unclear. Here, we report a PIB double-gyre that moves with the PIIS calving front and hypothesise that it controls ocean heat input towards PIIS. Glacial melt generates cyclonic and anticyclonic gyres near and off PIIS, and meltwater outflows converge into the anticyclonic gyre with a deep-convex-downward thermocline. The double-gyre migrated eastward as the calving front retreated, placing the anticyclonic gyre over a shallow seafloor ridge, reducing the ocean heat input towards PIIS. Reconfigurations of meltwater-driven gyres associated with moving ice boundaries might be crucial in modulating ocean heat delivery to glacial ice.

[1] School of Earth System Sciences, Kyungpook National University, 80, Daehak-ro, Buk-gu, Daegu 41566, Republic of Korea. [2] Division of Glacial Environment Research, Korea Polar Research Institute, Incheon 21990, Republic of Korea. [3] School of Earth and Environmental Sciences/Research Institute of Oceanography, Seoul National University, Gwanak-gu, Seoul 08826, Republic of Korea. [4] School of Environmental Sciences, University of East Anglia, Norwich, Norfolk NF4 7TJ, UK. [5] Scottish Oceans Institute, University of St Andrews, Andrews, Fife KY16 8LB, UK. [6] Department of Oceanography, Pusan National University, Geumjeong-gu, Busan 46241, Republic of Korea. [7] Department of Geography and Environmental Sciences, Northumbria University, Newcastle-upon-Tyne NE1 8ST, UK. [8] British Antarctic Survey, Cambridge CB3 0ET, UK. [9] Department of Earth and Atmospheric Sciences, University of Houston, Houston, TX 77004, USA. ✉email: wonsang@kopri.re.kr; namsh@snu.ac.kr

Antarctic ice shelves buttress the ice sheet and restrain the speed of the ice flow, dampening ice discharge to the ocean[1–3] and the associated sea-level rise. Many West Antarctic glaciers have recently been losing mass due to ice shelf thinning[4–6] and rapid grounding line retreat[7], thereby seriously threatening their stability. The heat transported by the relatively warm modified Circumpolar Deep Water (mCDW)[8–11] is the major heat source melting the West Antarctic ice shelves. Pine Island Ice Shelf (PIIS) is among the most rapidly melting ice shelves, whose feeding glacier is responsible for ~40% of the net ice mass loss from West Antarctica[6].

The basal melting rate of West Antarctic ice shelves varies owing to the variability in oceanic forcing at multiple timescales[12–14]. The variations in PIIS melting have been observed over weekly to decadal timescales[9,12,13,15,16], under the influence of ocean heat fluxes at the continental shelf break[11] and in Pine Island Bay (PIB)[12,16]. The thermocline depth in front of PIIS is considered to determine the heat transport into the ice shelf cavity (i.e. ocean below the ice shelf) by adjusting the mCDW thickness[8,15,16]. Although continental shelf break processes dominate the heat influx from the Southern Ocean[11], local (within PIB) sea-ice production and associated salt gain and heat loss have been reported as the forcings that deepen the thermocline depth at seasonal timescales[15,16]. In addition, frictional response to local wind forcing in front of PIIS has been proposed to dominate short-term variability in ocean heat flux[12].

The PIIS front has retreated rapidly since 2015, thereby changing the geographic boundary context of PIB[17–19]. Thus, the oceanic area in front of PIIS lengthened and narrowed (Fig. 1a), paving the way for potential ocean heat redistributions. Within the context of PIB, the area where the cyclonic gyre was in 2009[20] is relatively shallow (~900 m) (Fig. 1a). In 2009, upward convex of the mCDW layer (i.e. relatively shallow thermocline depth) allowed for a relatively large amount of heat to fill the basin in front of PIIS. A retreat of the ice front of ~30 km by 2020 modified this scenario (see a schematic figure below). In the following, we investigate the ocean circulation changes that followed the change in ice geometry and the subsequent changes in heat redistribution processes.

## Results

**Double-gyre in PIB.** To identify the influences of PIIS front retreat on ocean circulations in PIB, hydrographic profiles were obtained in PIB during January–February 2020 (Fig. 1a; Methods). The 2020 observations show two counter-rotating gyres (Fig. 1b, c). At the region where the cyclonic gyre was observed in 2009[20], an anticyclonic gyre with a radius of ~16 km was identified in 2020 (Fig. 1b, c). Its centre was located at ~74.84°S/102.80°W (Fig. 1b, c), and the volume transport circulating in the upper 700 m reached ~0.41 Sv. The cyclonic gyre typically observed in front of PIIS[20] (Fig. 1a, Supplementary Figs. 1 and 2) was also detected in 2020. However, in 2020, it was located at ~74.95°S/101.45°W with a radius of ~21 km in the newly exposed oceanic area following the PIIS front retreat (Fig. 1a, b). The cyclonic gyre observed in 2020 had a smaller radius than that in 2009 by 4 km (~25 km in 2009)[20] and circulated ~1.27 Sv in the upper 700 m in PIB, which was ~15% less than that observed in 2009 (~1.50 Sv)[20].

The counter-rotating double-gyre observed in 2020 can also be identified in 2009 and 2014 in shipborne Acoustic Doppler Current Profiler data and feature-tracked sea-ice motion vectors derived from satellite imagery (Fig. 1a, Supplementary Figs. 1 and 2b). However, only the cyclonic gyre and associated thermocline displacements (e.g. upward in the centre) in front of PIIS have been the focus of the previous studies[9,16,20,21]. Due to the PIIS front retreat, the double-gyre moved southeastward by more than

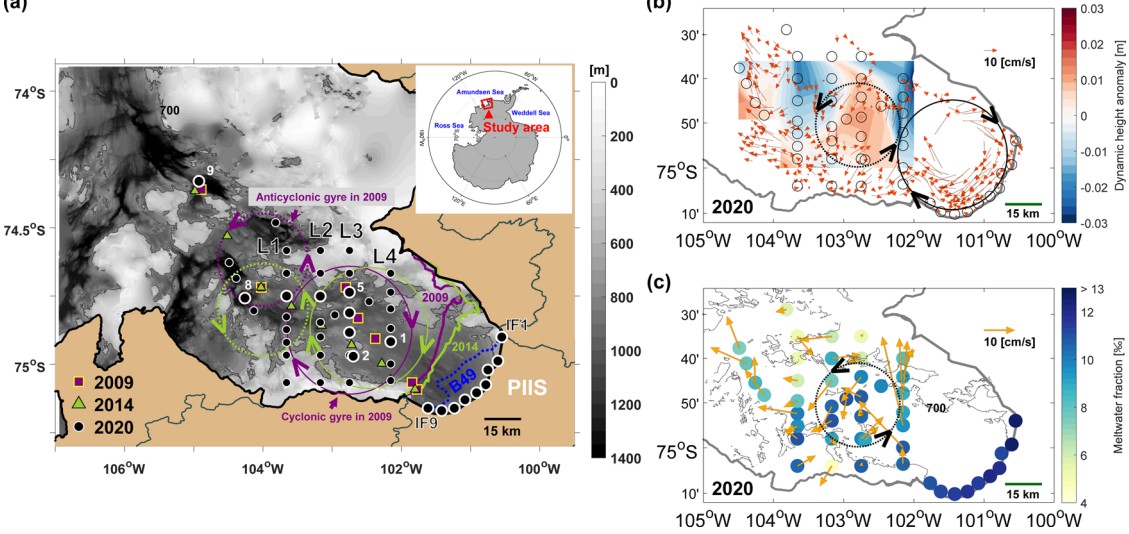

**Fig. 1 Circulation in PIB in 2020. a** Hydrographic observation map near PIIS from January–February 2020 overlaid on a local high-resolution bathymetry chart[28,48]. White-rimmed black circles represent the positions of 2020 Conductivity-Temperature-Depth (CTD) data presented in Fig. 2. The selected CTD stations are numbered in an order of distance from PIIS. L1-L4 and IF indicate CTD observational lines aligned meridionally across PIB and along PIIS, respectively. Solid purple and green lines represent the PIIS front positions in 2009 and 2014, respectively. The solid black line indicates a new ice shelf front after the calving of the 'B49' iceberg (dashed blue line) from PIIS on 9 February 2020. The large solid and dotted purple circles (green circles) with arrows indicate the approximate extents of the cyclonic and anticyclonic gyres observed in 2009[20] (2014), respectively. These extents of gyres are referred to from Supplementary Fig. 1. Dark grey contour denotes the grounding line. **b** Red-faced arrows denote the ocean currents averaged over depth ranging from 30 to 300 m based on Ship-based Acoustic Doppler Profiler (SADCP) data collected in 2020, gridded into ~3 × 3 km horizontal boxes. Shading represents the dynamic height anomaly (Methods). The large solid (dotted) black circle with arrows denotes the approximate size and position of the cyclonic (anticyclonic) gyre in 2020. Small black circles denote the positions of CTD stations. **c** Orange-faced arrows indicate the ocean currents averaged over depth ranging from 30 to 300 m based on the Lowered Acoustic Doppler Current Profiler (LADCP) data. Colour-filled circles indicate the vertically averaged meltwater fraction above the mCDW layer.

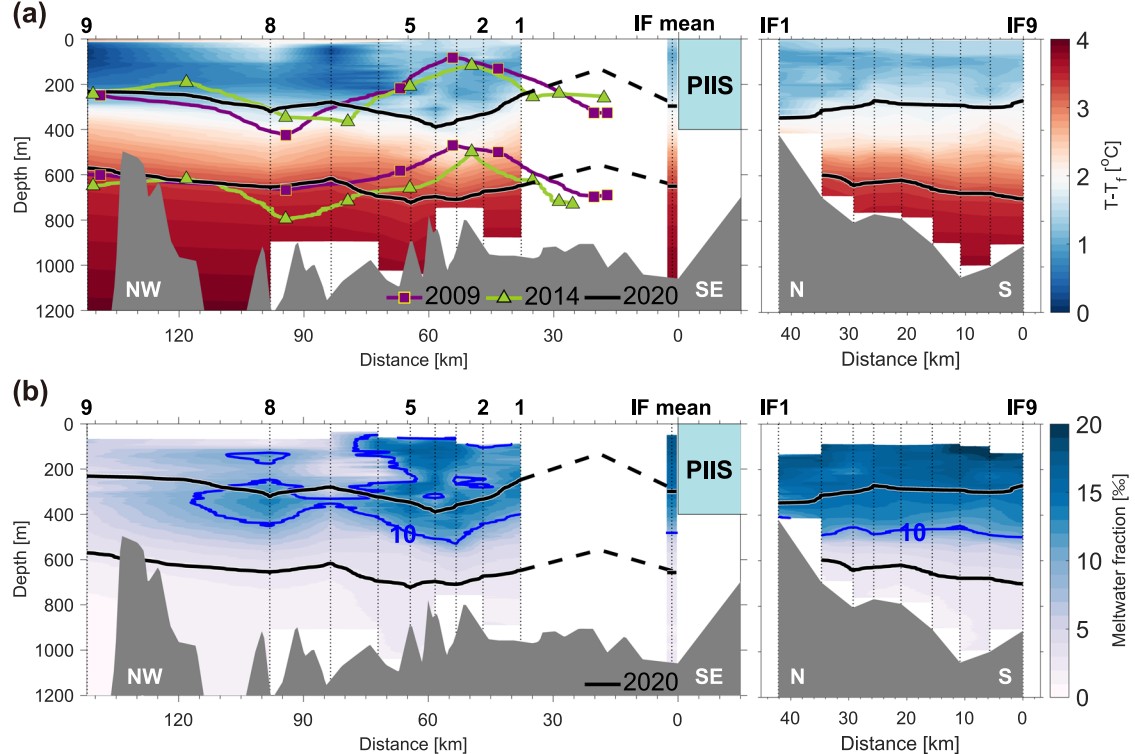

**Fig. 2 PIB hydrography in 2020. a** Vertical distribution of the temperature above the in situ freezing point ($T_f$) across PIB (left panel) and along the PIIS front (right panel, IF line) in 2020. The 27.47 kg/m³ and 27.75 kg/m³ (potential density) isopycnals in 2009, 2014 and 2020 are indicated by solid purple, green and black lines, respectively. Dashed black lines denote the simulated depths of the 27.47 kg/m³ and 27.75 kg/m³ isopycnals in 2020 (Methods). **b** Same as in **a**, but this panel shows the meltwater fraction in 2020. Blue contours indicate a 10‰ meltwater fraction. Values at depths with unreliable meltwater fractions were excluded (Methods).

20 km in 2020 compared with 2009 and 2014; that is, the location of the western limb of the cyclonic gyre in 2020 matches that of the eastern limb of the cyclonic gyre in 2009 (Fig. 1a, b).

Coinciding with the double-gyre migration following the PIIS front retreat (Fig. 1a, b), density structures were shifted further to the east in 2020 than those in 2009 and 2014 (Fig. 2a). An eastward vertical cross-section passing through the double-gyre centre was characterised by convex downward and upward isopycnals through the entire water column, represented by potential densities of 27.47 and 27.75 kg/m³ (Fig. 2a). The shapes of the isopycnals were similar to those observed in 2009[8] and 2014[21,22], although the slope of the density gradient in 2020 was slightly gentler than that in 2009 due to a relatively weaker gyre transport (Fig. 2a; Methods). In summary, in situ observations have allowed identification of a complex double-gyre pair in PIB and documented its relocation tracking the PIIS front retreat, resulting in the anticyclonic gyre seating over a shallower trough (<1000 m) off PIIS in 2020 (Figs. 1a, b, 2a, and Supplementary Fig. 3; see a schematic figure below).

Meltwater-rich glacially modified water (GMW) that buoyantly flows out from the ice cavity is a driver for cyclonic gyre formation in PIB, as it imparts cyclonic vorticity[23,24]. Numerical simulations that explicitly resolve the cavity and open sea circulation in a 'Pine Island Glacier-like' domain, in which melting is the only potential vorticity (PV) source in the vicinity of the ice shelf front, produce a gyre-train. (Supplementary Fig. 4; Methods). In such an idealised domain, only the first few gyres in the gyre-train can be considered realistic features as other PV sources (other than the melt) are expected to play an ever-increasing role with increasing distance from PIIS. Nevertheless, these simulations support the hypothesis that the double-gyre in

front of the PIIS (Fig. 1a, b) may have indeed been formed via changes in the PV input by glacial melt only. In the simulation, the horizontal extent of a counter-rotating double-vortex in front of the PIIS was similar to that of the 2020 observation (Fig. 1b). However, the volume transport of the cyclonic gyre is considerably smaller (~0.7 Sv) than was observed in 2020, perhaps indicating that other PV sources, which are not included in the simulation (most notably a local wind stress curl[20]) also strengthen the cyclonic gyre.

**Glacial meltwater circulation.** The GMW flows westward along the PIIS front and southern coast[22,23,25] according to the buoyant coastal plume theory[26] applied to the Southern Hemisphere. After the separation of the GMW from the southern coast, GMW is entrained and circulates in the double-gyre. According to the 2020 observations, this can be identified by the relatively high meltwater fraction above the mCDW layer (potential density > 27.75 kg/m³) along the ice front (>10‰) and the interface between the western rim of the cyclonic gyre and eastern rim of the anticyclonic gyre (>7‰) (Fig. 1c). Meltwater fraction appeared even higher (>10‰) within the anticyclonic gyre than at the gyre rim (Figs. 1c and 2b), which might be consistent with near-steady GMW transport towards the anticyclonic gyre centre due to horizontal convergence and downwelling above 500 m. Inversely, the meltwater fraction was relatively small within the cyclonic gyre owing to cyclonic gyre-induced upwelling and horizontal divergence[22].

Areas with high meltwater fraction within the anticyclonic gyre were characterised by positive dynamic height anomalies (Fig. 1b, c), because GMW with a 5‰ meltwater fraction was ~0.05 kg/m³

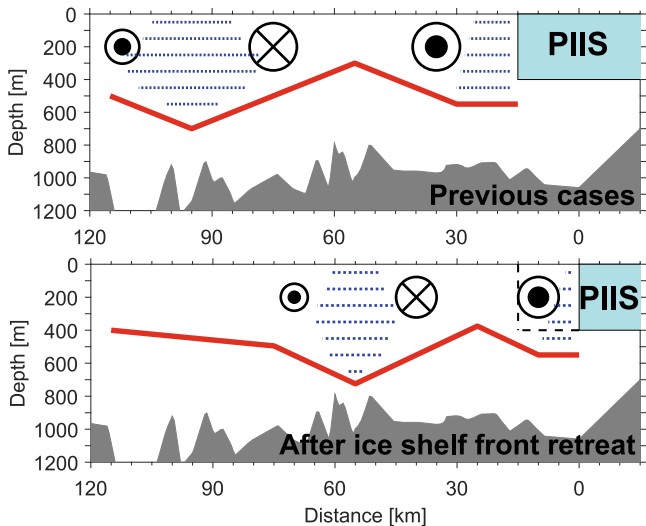

**Fig. 3 Schematic figure representing the location of the double-gyre and related meltwater distribution in PIB.** The upper and lower panels indicate oceanic conditions before (previous cases) and after the ice shelf front retreat, respectively (not to scale). The dashed black line in the lower panel indicates the ice shelf front position in the previous cases before the retreat. Dotted blue lines represent the water column with a high meltwater content. The thick solid red line shows the approximate variation in the isopycnals (also thermocline) associated with the meltwater distributions influenced by the counter-rotating double gyres. The circles with crosses and dots denote the ocean flows into and out of the plane of the diagram, respectively; their sizes indicate the relative flow speed and volume transport.

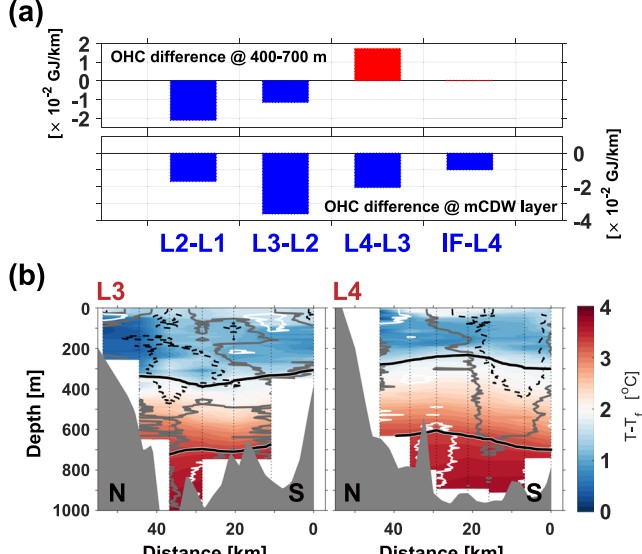

**Fig. 4 Ocean heat content for each north–south section. a** Bar graphs denote the differences in ocean heat content (OHC) among the observational lines relative to the line distance in the depth layer of 400–700 m (upper panel) and mCDW layer (lower panel) (Supplementary Table 1). **b** Vertical section of the temperature above the in situ freezing point along the L3 (left panel) and L4 (right panel; Fig. 1a) sections. Dashed black and solid white contours indicate a 5 cm/s outflow from the PIIS cavity and inflow to the PIIS cavity, respectively. Solid grey contours indicate zero velocity. Solid black contours denote the 27.47 kg/m$^3$ and 27.75 kg/m$^3$ (potential density) isopycnals in 2020.

lighter than the pure Winter Water (WW) produced during the previous winter[22] as observed in 2020 (Supplementary Fig. 5). The mean density difference between the centre and the edge was 0.04 kg/m$^3$ for the anticyclonic gyre and 0.08 kg/m$^3$ for the cyclonic gyre with respect to the reference depth of 700 m (Supplementary Fig. 6). These results indicate that when more GMW was converged into the anticyclonic gyre centre, that may displace the isopycnals farther downward at the centre. A schematic representation of the double-gyre relocation after the PIIS front retreat and the corresponding distribution of isopycnals and GMW is shown in Fig. 3.

**Heat redistribution by the ocean gyre.** In 2009 and 2014, the anticyclonic gyre with a high meltwater content was far from the PIIS front in both years (~80 km)[22], located in the open sea and covering a relatively deep part of the seabed (Figs. 1a, 3, and Supplementary Fig. 3). The eastward relocation of ocean gyres in 2020 tracking the PIIS front retreat placed the anticyclonic gyre instead at a relatively narrow region and over an ~200 m shallower constriction (Figs. 1b, 3, and Supplementary Fig. 3). Thus, the anticyclonic gyre mostly occupies the entrance where ocean heat delivers towards PIIS, and thins the lower ocean layer that contains and delivers most of the ocean heat content (OHC) to the ice. The relocation, therefore, opens the possibility that a double-gyre, at least partially created by glacial melt, may also be involved in modulating the delivery of oceanic heat to PIIS. To quantify heat redistribution by the double-gyre, the OHC and heat flux were analysed by assuming that our 2020 observations captured the bulk heat exchange between the inner and outer regions of PIB.

According to the 2020 observations, the OHC within 400–700 m depth (depth range between the ice draft and top of the ridge under PIIS[8,16]) and the mCDW layer thickness at the

entrance of PIB (L1 in Fig. 1a) were reduced by ~12% and ~14% in the anticyclonic gyre (L2 and L3 in Fig. 1a), respectively (Figs. 3 and 4a; Supplementary Table 1). The downward displacement of the isopycnals in the centre of the anticyclonic gyre caused a decrease in the mCDW layer thickness, resulting in an OHC reduction below 400 m (Figs. 3 and 4b). The substantial decrease in the OHC at the mCDW layer between L2 and L3 may also have been affected by the relatively shallow depth of the trough at the anticyclonic gyre location (Fig. 3 and Supplementary Fig. 3).

Furthermore, the heat flux at the mCDW layer in 2020 was considerably smaller at the centre of PIB compared to that in the 2009 case (before the recent PIIS front retreat). In 2009, convex upward isopycnals similar to those shown at L4 were observed near L3 (Fig. 4b and Supplementary Fig. 7). Due to the increased influence of the mCDW over this region, the mean heat flux at the mCDW layer was 1.45 TW ($1.45 \times 10^3$ GJ/s) in 2009 (Supplementary Fig. 7; Supplementary Tables 2 and 3; Methods). However, the mean heat flux was 0.61 TW ($0.61 \times 10^3$ GJ/s) in 2020 owing to the lower OHC and reduced velocity at the mCDW layer (Supplementary Tables 1, 2 and 3; Methods).

The OHC at 400–700 m increased at L4 (Fig. 1a) with the upward displacement of the isopycnals (Fig. 4b). However, the OHC was still smaller than that at L1 by ~4% (Fig. 4a; Supplementary Table 1). This might be because the increased OHC at L4 was partially offset by the substantially lower oceanic heat input via L3. The OHC at 400–700 m redistributed by the double-gyre finally delivered 3.41 GJ to the PIIS front (Supplementary Table 1). These results indicate that the heat input towards PIIS may be limited due to the ocean heat redistribution in the area covered by the relocated anticyclonic gyre.

## Discussion

Our observations demonstrated that the previously observed double-gyre relocated to the newly exposed region due to the recent PIIS front retreat (Figs. 1a, b, 2a and 3). The distribution of GMW in PIB was affected by the double-gyre circulation, and tended to be entrained within the secondary anticyclonic gyre at the centre of PIB in 2020 (Figs. 1c, 2b and 3). This circulation, together with shallow seafloor ridges beneath it, might play a role in reducing the available ocean heat input towards the ice shelf after the ice front retreat (Figs. 3 and 4). Thus, after the PIIS front retreated, negative meltwater feedback could be suggested for the basal melting of the ice shelf. This generated a feedback loop as follows, (i) increase in PIIS melting caused (ii) an increase in meltwater outflow, (iii) strengthening the anticyclonic gyre and increasing the meltwater accumulation within the gyre, (iv) decreasing the available OHC delivered towards PIIS by a deeper convex downward thermocline depth, (v) resulting in a reduction in the PIIS melt rate.

The meltwater flux from PIIS in 2020 (41.5 Gt/yr; Supplementary Fig. 8) was estimated to be approximately half of that in 2009 (79 Gt/yr)[8,9] and equivalent to that in 2012 (37 Gt/yr)[9] and 2014 (40 Gt/yr)[21]. The 2012–2014 period was characterised by consistently lower mCDW temperatures and melting rates than those in the 2009 period. Therefore, the 2012–2014 period was considered a relatively cool period[9,16,21]. Such a low meltwater flux in 2020 may result from step (v) in the negative meltwater feedback. However, we only partially elucidated the underlying mechanism using limited ship-based observations.

The negative meltwater feedback may control the ocean heat input and basal melting rate of PIIS on weekly or monthly scales[12] rather than longer timescales as step (v) may induce the deactivation of step (i) and (ii) in the negative meltwater feedback. In the future, the negative meltwater feedback may play a more crucial role in the basal melting of PIIS depending on PIIS front retreat distance as there are seafloor ridges (including a 400 m high seafloor ridge[9,27]) underneath PIIS, which are regarded as a modulator of ocean heat transport towards the grounding line[27]. Thus, the negative feedback mechanism warrants future investigations on ocean circulation in the PIB through year-round monitoring and/or numerical models that reflect geographic boundary changes and high-resolution bathymetry[28].

Enhanced meltwater input in a warming world may trigger a redirection of the coastal current leading to the frequent intrusion of warm water into the Antarctic continental shelf region, thereby accelerating ice shelf melting (e.g. Filchner-Ronne ice shelf, Weddell Sea)[29–31]. However, as proposed herein, a large amount of meltwater can occasionally produce negative feedback, which may moderate the basal melt rate depending on the frontal migration changes. Thus, this study has an important implication for other Antarctic ice shelves that the local ice-ocean interactions controlling the basal melting of ice shelves could be altered by geographic conditions linked to the frontal migration. The circum-Antarctic ice shelves have been retreating over the recent decades influenced by diverse environmental factors including, but not limited to ice front migration[32], yet our understanding of influences of these retreats on local ice shelf-ocean interactions remains poor. Understanding changes in the local ice shelf-ocean interaction caused by the retreat will be one of the key elements to improving capabilities for predicting future changes of ice shelves and global sea-level rise. Therefore, the Antarctic ice shelves that export a large amount of meltwater (such as the Thwaites Ice Shelf[33]) and are retreating rapidly[32] need to be continuously monitored based on comprehensive observations that consider mCDW pathways and detailed bathymetry. In addition, to improve the prediction of future sea-level rise due to Antarctic melting, ice shelf frontal migration and related meltwater distribution should be well incorporated into the models.

## Methods

**Hydrographic data**. We conducted full-depth conductivity-temperature-depth (CTD) and Lowered Acoustic Doppler Current Profiler (LADCP) casts at 29 stations along four meridional lines (L1, L2, L3 and L4 in Fig. 1a) from 29 January to 10 February 2020. This survey was conducted aboard the ice-breaking research vessel ARAON (Korea Polar Research Institute, KOPRI). The distance between each line (each station) was ~17 km (7 km). The first baroclinic Rossby radius of deformation, theoretically considered as the lower boundary of the ocean circulation radius[34], was estimated to be ~6 km in PIB. The spacing of the 2020 observations (~7 km) is comparable to the first baroclinic Rossby radius of deformation, indicating that our sampling space is sufficiently small to capture the ocean gyre circulations at horizontal scales of a few tens of kilometres in this area. The four meridional lines nearly cover the region where the cyclonic gyre was detected in 2009[20] (Fig. 1a). Full-depth CTD/LADCP casts were also conducted along the IF line (nine stations) aboard the ice-breaking research vessel Nathaniel B. Palmer (National Science Foundation, NSF; cruise NBP-2002), on 20 February 2020, after the 'B49' iceberg calving event. We only used full-depth CTD profiles along the IF line to estimate the meltwater flux from PIIS in 2020. All CTD profiles were measured using a SBE911 (Sea-Bird Electronic incorporated, US) with dual temperature and conductivity sensors. A down-looking Teledyne/RDI 300 kHz Workhorse-type ADCP attached to the CTD frame was used as a LADCP to measure the profiles of horizontal currents at a 5-m depth interval with an accuracy of typically ±0.5 cm/s (http://www.teledynemarine.com/).

The CTD data from both surveys were processed using the method recommended by Sea-Bird Electronics Incorporated[35]. All CTD profiles were arranged at a 1 m depth interval. The dissolved oxygen (DO) sensor data were calibrated using the following equation:

$$\text{Corrected DO (mL/L)} = (\text{Observed DO from the sensor (mL/L)} + 0.1996)/1.0380 \tag{1}$$

Equation (1) is based on the linear regression between the DO measured in 40 bottle samples using the Winker method and DO sensor data from four stations in PIB (not shown). We used two DO sensors with calibration dates of 4 April 2019 and 1 June 2019. The $R^2$ value for the correlation between the bottle samples and sensor values was 0.999.

The LADCP data were processed using the standard method[36], and de-tiding was not applied to the LADCP data because the observed current velocities were significantly stronger than the tidal velocities (<1 cm/s) in PIB[20]. The horizontal currents in PIB were also observed by ship-based ADCP (SADCP) from 29 January to 25 February 2020. The NBP had two Teledyne RDI ADCPs[37], both Ocean Surveyor models (phased array) operating at 75 kHz in narrow and broadband modes and at 38 kHz in narrowband mode. The lower-frequency ADCP could reach 1000 m in good conditions. However, the range was typically less in the PIB environment. A thick ice-protection window impacted the high-frequency ADCP. Therefore, it only reached 100–150 m in broadband mode and 400–450 m in narrowband mode. The University of Hawaii Data Acquisition System (UHDAS) combined ADCP and navigational data streams and used Common Ocean Data Access System (CODAS) processing to incrementally build a dataset of averaged (15 min) edited ocean velocities for each ADCP and ping type specified. The SADCP data, which had percent-good (percentage of available pings in an ensemble) values lower than 90%, were removed from this analysis. We used the horizontal ocean currents obtained by averaging the currents observed at three frequencies. In addition, historical SADCP data observed in 2009[20] and 2014 were also used to identify horizontal ocean circulations in PIB (Supplementary Fig. 1).

**Ice shelf front data**. The PIIS front in 2009 and 2014 was manually digitised from Moderate Resolution Imaging Spectroradiometer (MODIS) Antarctic ice shelf image[38] obtained on 30 January 2009 and 5 February 2014, respectively. The coastlines on 2 February and 8 March 2020 were also manually digitised based on sentinel-1 synthetic-aperture radar (SAR) images obtained on the same dates.

**Sea-ice motion**. Horizontal ocean surface currents can be inferred by sea-ice motion derived from MODIS images (Supplementary Fig. 2). First, we selected two MODIS images obtained at a short interval of ~100 min. Second, the travel distances of the sea-ice pixels were estimated using an optical flow technique[39]. Finally, the horizontal ocean surface current vectors were calculated by dividing the travel distance by the time difference between the two images. Considering that sea ice drifted owing to both the surface ocean current and wind drag, we selected the images obtained when the wind was not strong (<5 m/s) to estimate the sea-ice motion dominantly forced by the ocean currents. Considering the 250 m pixel size of the MODIS image and the time interval of ~100 min, the resolution of the sea-ice motion estimation was <4 m/s with the sub-pixel resolution ability of the optical flow technique. The directional accuracy of the optical flow technique was <4° for a synthetic image sequence[39]. The estimated sea-ice motion at a 250 m resolution was re-gridded at an ~4 km resolution, as shown in

Supplementary Fig. 2. Based on the estimated sea-ice motion, a cyclonic gyre occurred in front of PIIS in late 2012 (late 2019), and a double-gyre was observed in front of PIIS in late 2013.

**Meltwater fraction and meltwater flux.** The summer water column in front of PIIS primarily comprises Winter Water (WW), modified Circumpolar Deep Water (mCDW) and meltwater from PIIS (Supplementary Fig. 1c). Meltwater outflow from the PIIS cavity is added to the mixture of mCDW and WW; therefore, the meltwater fraction ($\varphi$) can be estimated using three independent tracers (i.e. potential temperature: PT; salinity: S; and DO measured from the CTD data):

$$\psi_{mix}^{2,1} = (\chi_{mix}^2 - \chi_{CDW}^2) - (\chi_{mix}^1 - \chi_{CDW}^1)\left(\frac{\chi_{WW}^2 - \chi_{CDW}^2}{\chi_{WW}^1 - \chi_{CDW}^1}\right), \quad (2)$$

$$\psi_{melt}^{2,1} = (\chi_{melt}^2 - \chi_{CDW}^2) - (\chi_{melt}^1 - \chi_{CDW}^1)\left(\frac{\chi_{WW}^2 - \chi_{CDW}^2}{\chi_{WW}^1 - \chi_{CDW}^1}\right), \quad (3)$$

and

$$\varphi = \frac{\psi_{mix}^{2,1}}{\psi_{melt}^{2,1}}, \quad (4)$$

where $\chi$ indicates PT, S and DO and the subscripted text represents the water mass properties[8,40]. The mCDW and WW endpoints were selected based on observational data (mCDW: PT of 1.28 °C, S of 34.75 and DO of 4.28 mL/L; WW: PT of −1.87 °C, S of 34.11 and DO of 7.03 mL/L) and the typical values of PT, S and DO for the ice water were approximately −90.75 °C, 0 and 28.46 mL/L, respectively[41] (Supplementary Fig. 5). The near-surface tracer values were considered to be partially contaminated by air–sea interactions and were thus excluded from this analysis (Fig. 2b; Supplementary Fig. 5).

The IF line was observed after the calving of the 'B49' iceberg from PIIS such that the CTD profiles of the IF line cover the region of the PIIS front at the 5 km horizontal scale. To estimate the meltwater flux in 2020, geostrophic velocities perpendicular to the IF line were calculated following the Thermodynamic Equation of Seawater – 2010 (TEOS-10)[42] using the depth of the 5‰ meltwater fraction as the reference level[8] (Supplementary Fig. 8). Subsequently, the velocities were adjusted using the tracer budget conservation method[8,43]. Finally, the meltwater flux was estimated from the difference between the volume transport of the inflow and outflow perpendicular to the line (mSv = $10^3$ m$^3$/s). Layers with unreliable meltwater fractions were excluded from this calculation (Supplementary Fig. 8). In 2020, the outflow volume transport was 1.32 mSv larger than that of the inflow. The total volume transport of 1.32 mSv can be converted into 41.5 Gt/yr by multiplying the value with the water density.

During the geostrophic velocity calculation, dynamic height anomalies with respect to 500 m were also estimated following TEOS-10[42] using CTD data from the PIB. The dynamic height anomaly indicates the geostrophic stream function. The anomalies in Fig. 1b indicate the dynamic height anomalies obtained by removing the spatially mean value over PIB.

**Simulation depths of isopycnals.** The depths of the 27.47 kg/m$^3$ and 27.75 kg/m$^3$ isopycnals at the location of the cyclonic gyre were simulated based on the linear relation between the density gradient and volume transport. The volume transport of the cyclonic gyre in 2009 was 1.5 Sv[20], and the slope of the concave 27.47 kg/m$^3$ (27.75 kg/m$^3$) isopycnal was ~7.2 m/km (5.7 m/km) (Fig. 2a). The average velocity from the surface to 700 m along the southern limb of the cyclonic gyre was ~0.086 m/s (Fig. 1b); therefore, the gyre volume transport was ~1.27 Sv (~0.086 m/s × 700 m × 21,000 m). As the volume transport of the cyclonic gyre in 2020 was ~15% smaller than that in 2009[20], the slopes of the concave 27.47 kg/m$^3$ and 27.75 kg/m$^3$ isopycnals in 2020 were 15% flatter than those in 2009 (6.1 and 4.8 m/km, respectively). Based on these assumptions, the 27.47 kg/m$^3$ and 27.75 kg/m$^3$ isopycnals were simulated from the western limb of the cyclonic gyre to the PIIS front (station #1 in Fig. 1a; Fig. 2).

**Ocean heat content (OHC).** The OHC was calculated using the temperature above the in situ freezing point in the 400–700 m depth range and the mCDW layer. The 400–700 m layer indicates the depth range between the ice draft and the top of the ridge beneath PIIS[4]. The OHC was estimated using the following equation:

$$H = \int_{Z_1}^{Z_2} \rho C_p (T - T_f) dZ, \quad (5)$$

where T is the temperature, $T_f$ is the in situ freezing temperature, $Z_1$ is 400 m for 400–700 m OHC and upper boundary of the mCDW for OHC at the mCDW layer, $Z_2$ is 700 m for 400–700 m OHC and bottom depth for OHC at the mCDW layer, $\rho$ is the ocean density (kg/m$^3$) and $C_p$ is the ocean heat capacity (J/kgK). The in situ freezing point was estimated following TEOS-10[42]. Both $\rho$ and $C_p$ were calculated based on T and S. The ocean heat content estimated based on shallow stations (<550 m) cannot represent the values in the 400–700 m layer; therefore, they were excluded from the estimation of the mean value for each line.

The average OHC was calculated for each line using the weighted arithmetic mean. The weight parameter for each value was determined using the number of data points used for the OHC calculation for each station (Supplementary Tables 4 and 5):

$$\bar{H} = \frac{\sum\limits_{i=1}^{n}(W_i \times H_i)}{\sum\limits_{i=1}^{n} W_i}, \quad (6)$$

where $W_i = N_i / \sum_{i=1}^{n} N_i$ ($\sum_{i=1}^{n} W_i = 1$), i is a station number from 1 to n and N is the number of data points used for the OHC calculation for each station. The changes in the OHC in the mCDW layer relative to the line distance shown in Fig. 4a were estimated using the differences in the mCDW OHC at each line (Supplementary Table 1).

**Heat flux at the mCDW layer.** The heat flux at the mCDW layer was calculated by multiplying the OHC by the mean velocity at the mCDW layer of each station:

$$\bar{U} = \frac{1}{Z_2 - Z_1} \int_{Z_1}^{Z_2} U dZ, \quad (7)$$

$$HT = \bar{U} \times \int_{Z_1}^{Z_2} \rho C_p (T - T_f) \times D dZ, \quad (8)$$

where $Z_1$ is the upper boundary of the mCDW layer, $Z_2$ is the bottom depth, U is the zonal velocity rotated by 5° (direction towards PIIS; Supplementary Fig. 7a) and D is the horizontal distance covered by a station (approximately the distance between lines). When the zonal velocity (unrotated case) or zonal velocity rotated by 10° (Supplementary Fig. 7a) was used, the mean velocity and heat flux were changed. However, the differences among them are considerably smaller than the difference between the 2009 case and the 2020 case (L3) (Supplementary Tables 2 and 3). The mean heat flux at the mCDW layer of the 2009 section and L3 were calculated using the method used for the OHC calculation. At station number 6 of L3 (Supplementary Tables 1, 2, 3, 4 and 5), we performed the CTD cast twice on 4 February and 10 February 2020. In this study, we used the CTD and LADCP data obtained on 10 February as representative profiles at station number 6 of L3. When we used the data observed on 4 February, the main results were almost unchanged (Supplementary Tables 1, 2 and 3).

**MITgcm model description.** We simulated the impact of meltwater using the Massachusetts Institute of Technology general circulation model (MITgcm)[44], which includes a static representation of ice shelves[45]. Using MITgcm, we explicitly resolved the cavity circulation in a 'Pine Island Glacier-like' domain (Supplementary Fig. 4); this domain is similar to the ISOMIP domain[46], albeit doubled in length, and with a section of the ice shelf removed so that the ice shelf in the simulation is similar in size to PIIS (length 65 km and width 40 km). The ocean is restored to the ISOMIP 'warm' conditions at the western wall (X = 1000 km), and the other two walls are rigid boundaries. The restoring timescale varies linearly from 1 h at the grid adjacent to the boundary to 1/2 day at the fifth grid cell inside the domain. In this simulation, the only sources of PV are the meltwater and the restoring boundary; using a long domain, with the restoring boundary located far (~800 km) from the ice shelf, we could ensure that the PV near the ice front is dominated by meltwater alone.

In the MITgcm, melting is parameterised using the three equation formulations with velocity-dependent transfer coefficients[47]. We used the standard drag coefficient of $2.5 \times 10^{-3}$ for the ice shelf in the momentum balance adjacent to the ice shelf, while the drag coefficient that enters the three equation formulations is calibrated to $1.0 \times 10^{-3}$ to ensure that the total melt flux in the simulation (53.8 Gt/y; 1.71 mSv) closely matches the estimated observed value in 2020 (41.5 Gt/yr; 1.32 mSv). The simulation was performed for 70 model years from an initial state where the domain is entirely filled with cold (−1 °C) water, and reached a steady-state after 30 model years or so. In the Supplementary, we only showed the simulation result averaged over the final 2 model years (Supplementary Fig. 4).

## Data availability

Raw data obtained from the ARAON survey in 2020 are available at the Korea Polar Data Center (CTD; LADCP). The CTD and SADCP data from the RV Nathaniel B Palmer survey in 2020 are publicly available at the British Oceanographic Data Centre (www.bodc.ac.uk). Historical CTD data obtained for Pine Island Bay in 2009 and 2014, which have been used in numerous studies[8,9,21,22,25], are also available at the British Oceanographic Data Centre (https://bodc.ac.uk).

## Code availability

The simulations were performed using MITgcm at checkpoint c67u, which is publicly accessibly at http://mitgcm.org/public/source_code.html. Files and code used to drive MITgcm, and simulation data and code used to produce Supplementary Fig. 4 are available at https://github.com/alextbradley/PIB-gyre-sim.

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

## Acknowledgements

This study was sponsored by a research grant from the Korean Ministry of Oceans and Fisheries (KIMST20190361; PM21020) and supported by the National Science Foundation and Natural Environment Research Council (NERC: Grants NE/S006419/1 and NE/S006591/1) for the TARSAN and the THOR projects, components of the International Thwaites Glacier Collaboration (ITGC). ITGC Contribution No. ITGC-061. The numerical simulation was carried out on ARCHER2, the U.K. national HPC facility (http://archer2.ac.uk/).

## Author contributions

S.-T.Y., W.S.L. and S.N. designed the study and prepared the first draft. S.-T.Y. performed most of data processing. S.-T.Y., W.S.L., S.N., S.Y., I.L. and Y.C. contributed to oceanographic data collection from the LIONESS-TG cruise. K.H., L.B., Y.Z., R.L. and J.W. collected oceanographic data from the ITGC cruise. C.-K.L. claims responsibility for Supplementary Fig. 2. A.T.B and P.D. performed the simulation using MITgcm and contributed to analysing the simulation result. S.-T.Y. and E.K.J. interpreted atmospheric forcing. S.-T.Y., W.S.L., S.N., K.H., Y.Z., A.J. and P.D. contributed to the interpretation of the oceanographic data and the results. All authors have read and provided comments on the manuscript. W.S.L. and S.N. provided guidance and supervised the study.

## Competing interests

The authors declare no competing interests.

**Additional information**

