## [Peer Review File · Nature Communications]

Ice front retreat reconfigures meltwater-driven gyres
modulating ocean heat delivery to an Antarctic ice shelfReviewers' Comments:

Reviewer #1:

Remarks to the Author:

Review on "Ice front retreat reconfigures meltwater-driven gyres modulating ocean heat delivery to an Antarctic Ice shelf" by Seung-Tae Yoon et al. (2021)

Key results:

The authors present observations of current and hydrography in the Pine Island Bay region which show the presence of a cyclonic/anticyclonic gyre pair in the Pine Island Bay. The cyclonic gyre is located close to the ice shelf front, and the anticyclonic gyre is located a little further west. Based on data from 2009, 2012, and 2020, the authors show that the gyre pair has shifted eastward following the Pine Island ice shelf front retreat. The gyre pair leads to sloping isopycnals in the center of the gyres. The combination of these sloping isopycnals and the changes in the water column depth caused by the relocation of the gyre pair alters the gyres' role in delivering heat to the ice shelf base. The authors suggest a negative ice shelf melting feedback loop caused by the gyre pair. An initial increase in ice shelf melting will be dampened through increased glacial meltwater discharge, relocation of the gyre pair, and a reduced oceanic heat supply.

Validity:

The authors present a convincing story backed up by several different data set. The presence of the gyre pair is documented by both ship-based hydrographic and current data and surface currents from sea ice motion in the MODIS data set. The interpretations and conclusions include valid quantification of volume transports, heat content, and meltwater fractions and fluxes.

Significance:

The manuscript builds on relevant literature from the Pine Island Bay region. The cyclonic gyre is given attention in earlier work. This comprehensive study of the gyre pair and their relocation from 2009-2020 expands and builds on this works and provides a detailed discussion of the gyres' role in supplying heat to the Pine Island ice shelf base. However, the authors could improve the significance of the study by including a statement of the relevance of this study to other regions in Antarctica. Also, it would be interesting to include a statement about the expected future of the Pine Island Bay region. For example, how will the bathymetric conditions influence the heat supply if the gyre pair is shifted even further east?

Data and methodology:

The study relies on data from commonly used instruments produced by well-known companies, and the data processing is based on recommended methods. However, I could not find a description of the instrument type of the LADCP, so this would be useful to include. It would also be good to include the instruments' accuracy and precision in the methods section. The methods are otherwise well described and are appropriate for this study.

Analytical approach:

The analytical approach holds a high standard. The evidence of a gyre pair is strong and based on identifications from several data sets, including idealized numerical simulations. In addition, the study includes quantifications of relevant parameters that support the main conclusions of the manuscript.

Suggested improvements:

The selection of material and the message in the manuscript is well conveyed to the reader. I have no major issues to present, but I would suggest some minor changes to the graphics that could improve the readability and be helpful for the general reader.

My main suggestion for improvements is to increase the paper's relevance by commenting on the possibility of transferring the results from this study to other regions. See further comments under the section "significance."

Furthermore, I would encourage the authors to move the description and discussion of figure 4 to the

start of the manuscript. Figure 4 is very informative for a reader unfamiliar with the Pine Island Bay circulation and bathymetry. By placing this figure as the second figure and explain your model for the gyre pair, including the sloping isopycnals and the varying bathymetry, early in the manuscript, you could refer to this figure in the section on heat redistribution. Then the reader will more easily follow the main paper without having to read the extended material and figures to understand the main points.

Below follows a small selection of minor suggestions:

- 1) In Figure 1a, it would be helpful to also include the anticyclonic gyre location in 2009. Then the reader can more easily compare the sites from 2009 in fig 1a and 2020 in fig 1b. It would also be helpful to indicate the gyre circulation direction by an arrow on each circle.
- 2) It would be helpful to include legends or annotations to identify the isopycnals from different years in figure 2a and identify the ice shelf fronts in figure 1a.
- 3) The acronyms CTD, SADCP, and LADCP in the caption of figure 1 are not defined. They are relatively common in oceanography but should be defined in the text of the main paper.
- 4) Line 159: The sentence refers to figure 3b. It is not easy to see this result in figure 3b. Perhaps it would be helpful to link this statement to figure 4 instead?
- 5) Line 194-196: The sentence is incomplete. Please rewrite.
- 6) Line 196: The sentence starting here needs a grammar check.
- 7) Line 447: Method section – The definition of the mCDW density threshold would have been helpful to move to the main paper.

Clarity and context:

The manuscript is well written, and the text flows well while reading. The authors put the paper's topic in a scientific perspective and differentiate between their and previous findings. There is also a clear differentiation between results and the authors' further hypotheses/suggestions, and the data support the conclusions.

References:

The manuscript reference previous literature appropriately and includes references to manuscripts published during the past ten years.

Your expertise:

The tracking of sea ice motion from satellite images is outside the scope of my expertise.

Sincerely,
Dr. Kjersti Daae
Geophysical Institute
University of Bergen
Norway

Reviewer #2:

Remarks to the Author:

Review of "Ice front retreat reconfigures meltwater-driven gyres modulating ocean heat delivery to an Antarctic ice shelf" by Yoon, Lee and others.

Review by Dr Natalie Robinson, NIWA, NZ

This manuscript uses direct in-situ oceanographic observations to identify and probe a mechanism which may play a significant role in basal meltwater production from Antarctic ice shelves. Following retreat of the Pine Island Ice Shelf, a relocation of cyclonic/anticyclonic gyre pairs was observed, which, through adjusting the available heat content through interaction of stratification with local bathymetry, led to a net reduction in basal melting of the ice shelf. To my knowledge, this phenomenon has not previously been documented and is likely to be of immediate interest to the international community of researchers working in this field. I found the analysis to be an excellent

use of the available data, the manuscript coherent and generally well written, and the accompanying figures appropriate and useful. I have a few suggestions which I believe will improve clarity for the reader.

Major comment:

The analysis uses similar observational data collected across three separate years (2009, 2014, 2020). In some cases additional care in identifying which data were collected in which year would assist the reader. For example, from my reading I think that, in 2009, only the cyclonic gyre was directly observed, while the anticyclonic gyre was inferred from other supporting information (although not supported by the simulations?) But this may not be the case, and clarity on what was actually observed in which year would be helpful.

Minor comments:

NB: Care is required throughout in keeping tenses consistent. I have made suggestions below in every case I identified where past tense should be applied, but ultimately this is an editorial decision.

LL 38 'moves with' is sufficient for the purpose here. I think 'and tracks' might be more specific than can be justified by the temporal spacing of the observations.

LL 58 The statement 'under the influence of ocean heat fluxes' needs a reference.

LL 62 Southern Ocean should be capitalised.

LL 64-65 In general, the introductory material is quite high level and understandable by a fairly general audience. If 'Ekman pumping' is to be mentioned as a relevant phenomenon here, I think it needs just a few additional words of description.

LL 68 I think 'near PIIS' might be better expressed as 'of PIB'.

LL 72 Suggest exchanging '...ice front by approximately...' for 'ice front of approximately...' to avoid 2x 'by' in rapid succession.

LL 77-88 Melding 2x years of data into the same description and even sentence made this paragraph a bit confusing. Please take extra care to clearly identify which data and features were from which year. E.g. (LL 80-81) 'At the region where the cyclonic gyre developed in 2009, an anticyclonic gyre with a radius of ~16 km was identified in 2020.'

LL 80 - 81 '... the cyclonic gyre developed' should be '... a cyclonic gyre was observed'

LL 84 '... has also been...' should be '... was also...'

LL 84 'However,' should start a new sentence.

LL 84 'it is now located at' should be 'in 2020 was located at'

LL 85 'owing to' should be 'following', unless there are other data in support of causation.

LL 94 Suggest replacing 'towards PIIS' with 'South West'. Unless the gyre was actually closer to the ice front in 2020 (by 20 km) than it had been in 2009 and 2014. More clarity required.

LL 95-96 This point could be made clearer in the figures. My suggestion would be to include all cyclonic (solid line) and anticyclonic (dotted line) gyres for the three years on the same plot, with consistent colour-coding between figures.

LL 104 '... in-situ observations have allowed identification of...'

LL 110 'PIG' is used here for the first (and only?) time without expansion. I suggest writing 'Pine Island Glacier' out in full (with no abbreviation necessary).

LL 115-116 Again perhaps my understanding is not complete (as per major comment above), but from these lines I am left with the impression that the simulations showed formation of the double-gyre to be a result of ice front retreat. And therefore that only a single gyre may have been present prior to retreat. This seems inconsistent with previous (observational) material and some clarity would help.

LL 119-120 Suggest: "...considerably smaller (~0.7 SV) than was observed in 2009, (*) perhaps indicating that other PV sources which are not included..."

*Also not clear whether it would be 2009, 2014 or 2020 in the above sentence.

LL 129 Suggest: "Meltwater fraction appeared even higher..."

LL 145 - 149 This may be resolved with the improvement in clarity as suggested above, but the claim that 'the relocation of ocean gyres in 2020 tracking the PIIS front retreat...' seems at odds with the opening statement of the section that the anticyclonic gyre in 2009 and 2014 was far from the PIIS

front. Perhaps this means that the location of the anticyclonic gyre in 2009 and 2014 was far from the 2020 location of the PIIS front? Or perhaps it implies that in 2009 and 2014 the cyclonic and anticyclonic gyres were much further apart than in 2020, and the cyclonic (only) gyre did indeed relocate to 'track' the ice front retreat. But because the statement above refers to 'gyres' (rather than a single gyre) this doesn't seem right either.

LL 152 Addition of a couple of commas required. i.e. "... the possibility that a double-gyre, at least partially created by glacial melt, may also be..."

LL 156-159 It would be helpful to explicitly state the period over which the of OHC occurred. i.e. I think it was between 2009 and 2020, but can't be certain.

LL 172 'However,' starts a new sentence.

LL 180 ff Could adjust all tenses in the description of the negative feedback mechanism to be past tense (in line with the rest of the manuscript), but this is probably not necessary for just this section.

LL 194 Should this be 'low mCDW' or 'lower mCDW'?

LL 196 Should be '... in 2020 may result from...'

LL 199 Is there any basis in the literature for the suggested 'weekly or monthly' timescales. If there is, highlighting it would be really interesting and supportive here.

Fig. 1 It would aid clarity if the 4x subpanels were labelled (a), (b), (c), and (d). Alternatively the map of Antarctica could be more clearly incorporated as an inset of panel (a).

Fig 1 It might be really helpful if the colour-coding could be kept consistent across the figure – and even more so if it could be kept consistent across all figures. i.e. All site locations, gyres, and current meter observations from 2009 = purple; from 2014 = green; from 2020 = black. Currently 2020 is represented in Fig 1 with a combination of black/green/grey, so a potential source of confusion that could be easily remedied.

Fig 1(b) Which year were the SADCP data collected (could just add this to the caption).

LL 337 – 339 I just wanted to double-check that the logic follows here... Since the Rossby radius of deformation is smaller than the spacing of the observations, it's not immediately clear than the 'observational data can capture the ocean circulation...'

LL 363 'However,' to start the new sentence.

LL 369 'have' => 'had'

LL 476 'However,' to start the new sentence.

LL 495 '... located a far...' (i.e. delete 'a')

Extended Data Fig. 3 I think it would be really helpful to include all cyclonic and anticyclonic gyres on a single map, consistently colour-coded by year (perhaps this could be done in Fig.1 making ED Fig 3 redundant?). Suggest solid lines for cyclonic and dotted lines for anticyclonic. If some of these are from simulation only (i.e. not observational data) this should be included somehow (perhaps in the caption?)

Response to reviewers' comments

Reviewer #1

Review on "Ice front retreat reconfigures meltwater-driven gyres modulating ocean heat delivery to an Antarctic Ice shelf" by Seung-Tae Yoon *et al.* (2021)

1. Key results: The authors present observations of current and hydrography in the Pine Island Bay region which show the presence of a cyclonic/anticyclonic gyre pair in the Pine Island Bay. The cyclonic gyre is located close to the ice shelf front, and the anticyclonic gyre is located a little further west. Based on data from 2009, 2012, and 2020, the authors show that the gyre pair has shifted eastward following the Pine Island ice shelf front retreat. The gyre pair leads to sloping isopycnals in the center of the gyres. The combination of these sloping isopycnals and the changes in the water column depth caused by the relocation of the gyre pair alters the gyres' role in delivering heat to the ice shelf base. The authors suggest a negative ice shelf melting feedback loop caused by the gyre pair. An initial increase in ice shelf melting will be dampened through increased glacial meltwater discharge, relocation of the gyre pair, and a reduced oceanic heat supply.

We thank the reviewer for the excellent summary. We have made all the suggested changes as detailed below (blue).

2. Validity: The authors present a convincing story backed up by several different data sets. The presence of the gyre pair is documented by both ship-based hydrographic and current data and surface currents from sea ice motion in the MODIS data set. The interpretations and conclusions include valid quantification of volume transports, heat content, and meltwater fractions and fluxes.

The validity of this study is adequately described above.

3. Significance: The manuscript builds on relevant literature from the Pine Island Bay region. The cyclonic gyre is given attention in earlier work. This comprehensive study of the gyre pair and their relocation from 2009-2020 expands and builds on this work and provides a detailed discussion of the gyres' role in supplying heat to the Pine Island ice shelf base. However, the authors could improve the significance of the study by including a statement of the relevance of this study to other regions in Antarctica. Also, it would be interesting to include a statement about the expected future of the Pine Island Bay region. For example, how will the bathymetric conditions influence the heat supply if the gyre pair is shifted even further east?

*We thank the reviewer for giving valuable comments to better represent the significance of the study. The circum-Antarctic ice shelves have been retreating over the recent decades influenced by the variation in diverse environmental factors (Baumhoer *et al.* 2021). So, it is fair to expect that there may be similar cases to PIIS among the Antarctic ice shelves. We tried to search for other ice shelves where*

the dynamic factor we suggested in this manuscript (double-gyre pair in front of an ice shelf and bathymetric changes in relation to ice front migration) may play a significant role in modulating basal melting under ice shelves and mCDW inflow transporting to the grounding line. However, with lack of sufficient/robust knowledge from previous reports on such dynamics in other regions in Antarctica, it is difficult to link the process underlying the gyre's role in supplying heat to other ice shelf cavities. Thus, we chose not to include a statement of the relevance of this study to other ice shelves explicitly. Instead, we highlighted the important implications of this study on other Antarctic regions that local ice-ocean interactions controlling the basal melting of ice shelves could be altered due to gyre circulation and geographic condition relative to ice front via the frontal migration. We included this point in the Discussion of the revised manuscript. In addition, following the suggested comments, we inserted, in the Discussion, a statement on potential importance of seafloor ridges underneath the PIIS on future basal melting of PIIS and retreats of ice front and grounding line via the negative meltwater feedback process. Understanding changes in the local ice shelf-ocean interaction (including, but not limited to the ocean circulation patterns at mesoscale such as the double-gyre pair presented in this study, or scales finer than the resolution of general ocean circulation models) caused by the retreat may be one of the key elements to improving capabilities for predicting future changes of ice shelves, which support the necessity of continuous monitoring of the Antarctic ice shelves.

4. Data and methodology: The study relies on data from commonly used instruments produced by well-known companies, and the data processing is based on recommended methods. However, I could not find a description of the instrument type of the LADCP, so this would be useful to include. It would also be good to include the instruments' accuracy and precision in the methods section. The methods are otherwise well described and are appropriate for this study.

We have added, in the Methods of the revised manuscript, a description on the details of the LADCP and data collected using the instrument.

5. Analytical approach: The analytical approach holds a high standard. The evidence of a gyre pair is strong and based on identifications from several data sets, including idealized numerical simulations. In addition, the study includes quantifications of relevant parameters that support the main conclusions of the manuscript.

The analytical approach of this study is adequately described above.

6. Suggested improvements: The selection of material and the message in the manuscript is well conveyed to the reader. I have no major issues to present, but I would suggest some minor changes to the graphics that could improve the readability and be helpful for the general reader. My main suggestion for improvements is to increase the paper's relevance by commenting on the possibility of transferring the results from this study to other regions. See further comments under the section "significance." Furthermore, I would encourage the authors to move the description and discussion of figure 4 to the start of the manuscript. Figure 4 is very informative

for a reader unfamiliar with the Pine Island Bay circulation and bathymetry. By placing this figure as the second figure and explain your model for the gyre pair, including the sloping isopycnals and the varying bathymetry, early in the manuscript, you could refer to this figure in the section on heat redistribution. Then the reader will more easily follow the main paper without having to read the extended material and figures to understand the main points.

We thank the reviewer for suggestions to improve this paper. As responded above (the section “significance”), we highlighted the important implications of this study on other Antarctic regions regarding the gyre circulation and geographic condition related to ice front migration in the Discussion of the revised manuscript. We also revised the presentation by changing the order of figures, e.g., placing Figure 4 (of the original manuscript) before the descriptions on (and the figure relevant to; Figure 3 of the original manuscript or Figure 4 of the revised manuscript) ‘Heat redistribution by the ocean gyre’ in the Results of revised manuscript. Additionally, following the ‘Guide to formatting article’ of this journal, we added a statement ‘see a schematic figure below’ in the Introduction of the revised manuscript.

Below follows a small selection of minor suggestions:

7. 1) In Figure 1a, it would be helpful to also include the anticyclonic gyre location in 2009. Then the reader can more easily compare the sites from 2009 in fig 1a and 2020 in fig 1b. It would also be helpful to indicate the gyre circulation direction by an arrow on each circle.

Following these and other reviewer’s comments, we modified the Figure 1 to demonstrate approximate extents of cyclonic (solid circles) and anticyclonic (dotted circles) gyres observed in 2009 (purple) and 2014 (green) along with those observed in 2020 (black). In addition, arrows were added to indicate the gyre circulation direction in Figures 1a and 1b (see modified Figure 1 below).

Figure 1

8. 2) It would be helpful to include legends or annotations to identify the isopycnals from different years in figure 2a and identify the ice shelf fronts in figure 1a.

We included annotations and legends to Figures 1a and 2a in a consistent manner (using different colors and symbols for different years) as suggested.

9. 3) The acronyms CTD, SADC, and LADCP in the caption of figure 1 are not defined. They are relatively common in oceanography but should be defined in the text of the main paper.

The acronyms were fully described in the caption of Figure 1, consistently with the descriptions in the Methods of the revised manuscript.

10. 4) Line 159: The sentence refers to figure 3b. It is not easy to see this result in figure 3b. Perhaps it would be helpful to link this statement to figure 4 instead?

We additionally linked Figure 3 (of the revised manuscript) to the sentence in the revised manuscript as suggested.

11. 5) Line 194-196: The sentence is incomplete. Please rewrite.

We separated the sentence into two sentences for better readability.

12. 6) Line 196: The sentence starting here needs a grammar check.

The sentence was corrected.

13. 7) Line 447: Method section – The definition of the mCDW density threshold would have been helpful to move to the main paper.

Following the comments, we moved the definition of the mCDW density threshold to the main body.

Reviewer #2

Review of “Ice front retreat reconfigures meltwater-driven gyres modulating ocean heat delivery to an Antarctic ice shelf” by Yoon, Lee and others.

1. This manuscript uses direct in-situ oceanographic observations to identify and probe a mechanism which may play a significant role in basal meltwater production from Antarctic ice shelves. Following retreat of the Pine Island Ice Shelf, a relocation of cyclonic/anticyclonic gyre pairs was observed, which, through adjusting the available heat content through interaction of stratification with local bathymetry, led to a net reduction in basal melting of the ice shelf. To my knowledge, this phenomenon has not previously been documented and is likely to be of immediate interest to the international community of researchers working in this field. I found the analysis to be an excellent use of the available data, the manuscript coherent and generally well written, and the accompanying figures appropriate and useful. I have a few suggestions which I believe will improve clarity for the reader.

We thank the reviewer for suggestions that improved this paper significantly. The following text provides our point-by-point responses (blue) to the reviewer's comments (black).

Major comment:

2. The analysis uses similar observational data collected across three separate years (2009, 2014, 2020). In some cases additional care in identifying which data were collected in which year would assist the reader. For example, from my reading I think that, in 2009, only the cyclonic gyre was directly observed, while the anticyclonic gyre was inferred from other supporting information (although not supported by the simulations?) But this may not be the case, and clarity on what was actually observed in which year would be helpful.

Following the comments, we clarified on what was actually observed in which year, in the revised manuscript. We revised the texts to make the points below clearer.

Supplementary Fig. 1

As shown in Supplementary Fig. 1 of the revised manuscript, we identified a cyclonic/anticyclonic gyre pair in front of the Pine Island Ice Shelf (PIIS) directly observed from SADCP in (a) 2009 and (b) 2014. The double-gyre pair was also

confirmed from the cross-sectional structures of isopycnals across the Pine Island Bay (PIB) in 2009 and 2014 (convex upward and downward isopycnals shown in Figure 2a)—their presence have never been reported as previous studies (Thunrherr et al., 2014; Garabato et al. 2017) focused on the cyclonic gyre only. This is the first study reporting the double-gyre pair in the PIB based on both in-situ and remote sensing observations, and numerical simulations explain how it forms—changes in the potential vorticity input by glacial melt create the gyre pair in front of the PIIS.

Minor comments:

3. NB: Care is required throughout in keeping tenses consistent. I have made suggestions below in every case I identified where past tense should be applied, but ultimately this is an editorial decision.

We thank the reviewer for carefully assessing of our manuscript. We have completed all the suggested changes as follows.

4. LL 38 ‘moves with’ is sufficient for the purpose here. I think ‘and tracks’ might be more specific than can be justified by the temporal spacing of the observations.

We changed the wording as suggested.

5. LL 58 The statement ‘under the influence of ocean heat fluxes’ needs a reference.

We inserted relevant references: Thoma et al. 2008; Webber et al. 2017; Davis et al. 2018.

6. LL 62 Southern Ocean should be capitalised.

We capitalised the words.

7. LL 64-65 In general, the introductory material is quite high level and understandable by a fairly general audience. If ‘Ekman pumping’ is to be mentioned as a relevant phenomenon here, I think it needs just a few additional words of description.

We replaced ‘Ekman pumping processes’ with ‘frictional response to local wind forcing’ in the revised manuscript.

8. LL 68 I think ‘near PIIS’ might be better expressed as ‘of PIB’.

We changed the words as suggested.

9. LL 72 Suggest exchanging ‘...ice front by approximately...’ for ‘ice front of approximately...’ to avoid 2x ‘by’ in rapid succession.

We changed the words as suggested.

10. LL 77-88 Melding 2x years of data into the same description and even sentence made this paragraph a bit confusing. Please take extra care to clearly identify which data and features were from which year. E.g. (LL 80-81) 'At the region where the cyclonic gyre developed in 2009, an anticyclonic gyre with a radius of ~16 km was identified in 2020.'

We revised the paragraph to clarify which data and features were from which year as suggested.

11. LL 80 - 81 '... the cyclonic gyre developed' should be '... a cyclonic gyre was observed'

We changed the words as suggested.

12. LL 84 '... has also been...' should be '... was also...'

We changed the words as suggested.

13. LL 84 'However,' should start a new sentence.

We changed the sentence as suggested.

14. LL 84 'it is now located at' should be 'in 2020 was located at'

We changed the words as suggested.

15. LL 85 'owing to' should be 'following', unless there are other data in support of causation.

We changed the words as suggested.

16. LL 94 Suggest replacing 'towards PIIS' with 'South West'. Unless the gyre was actually closer to the ice front in 2020 (by 20 km) than it had been in 2009 and 2014. More clarity required.

We replaced 'towards PIIS' with 'southeastward' in the revised manuscript.

17. LL 95-96 This point could be made clearer in the figures. My suggestion would be to include all cyclonic (solid line) and anticyclonic (dotted line) gyres for the three years on the same plot, with consistent colour-coding between figures.

As suggested by this and other reviewers, we modified Figure 1, which may help the reader to compare locations of the double-gyre pair for different years effectively.

18. LL 104 '... in-situ observations have allowed identification of...'

We changed the words as suggested.

19. LL 110 'PIG' is used here for the first (and only?) time without expansion. I suggest writing 'Pine Island Glacier' out in full (with no abbreviation necessary).

We changed the words without using the abbreviation.

20. LL 115-116 Again perhaps my understanding is not complete (as per major comment above), but from these lines I am left with the impression that the simulations showed formation of the double-gyre to be a result of ice front retreat. And therefore that only a single gyre may have been present prior to retreat. This seems inconsistent with previous (observational) material and some clarity would help.

The sea ice motion tracked from the MODIS imagery and the pattern of surface current from SADC observations clearly show the presence of both gyres (not only the cyclonic gyre but also the anticyclonic gyre) in the PIB. The barotropic stream function derived from the idealized model simulations present that glacial meltwater alone could generate a 'gyre-train' in front of the PIIS. The message we try to deliver here is that the double-gyre system can be generated by the potential vorticity input by glacial melt (with no other source), and not that the observed system can be simulated realistically.

21. LL 119-120 Suggest: "...considerably smaller (~0.7 SV) than was observed in 2009, (*) perhaps indicating that other PV sources which are not included..." *Also not clear whether it would be 2009, 2014 or 2020 in the above sentence.

We changed the sentence for clarity to read: "The volume transport of the cyclonic gyre in the model simulation was considerably smaller (~ 0.7 Sv) than that was observed in 2020".

22. LL 129 Suggest: "Meltwater fraction appeared even higher..."

We changed the order of words as suggested.

23. LL 145 – 149 This may be resolved with the improvement in clarity as suggested above, but the claim that 'the relocation of ocean gyres in 2020 tracking the PIIS front retreat...' seems at odds with the opening statement of the section that the anticyclonic gyre in 2009 and 2014 was far from the PIIS front. Perhaps this means that the location of the anticyclonic gyre in 2009 and 2014 was far from the 2020 location of the PIIS front? Or perhaps it implies that in 2009 and 2014 the cyclonic and anticyclonic gyres were much further apart than in 2020, and the cyclonic (only) gyre did indeed relocate to 'track' the ice front retreat. But because the statement above refers to 'gyres' (rather than a single gyre) this doesn't seem right either.

In the revised manuscript, we changed the sentences to avoid the unnecessary confusion. The anticyclonic gyre with a high meltwater content was far distant from the Pine Island Ice Shelf (PIIS) front (> 80 km) in 2009 and 2014, which can be

observed in Figures 2 and 4 of the original manuscript (Figures 2 and 3 of the revised manuscript). However, the anticyclonic gyre was located closer to the PIIS front in 2020 (~ 50 km) due to the eastward relocation of ocean gyres (double-gyre) following the PIIS front retreat (Figures 2 and 4 of the original manuscript or Figures 2 and 3 of the revised manuscript).

24. LL 152 Addition of a couple of commas required. i.e. "... the possibility that a double gyre, at least partially created by glacial melt, may also be..."

We added commas in the revised manuscript.

25. LL 156-159 It would be helpful to explicitly state the period over which the of OHC occurred. i.e. I think it was between 2009 and 2020, but can't be certain.

In the revised manuscript, we clarified the year when the OHC reduction occurred at L2 and L3. The OHC reduction at L2 and L3 (corresponding to the anticyclonic gyre) was observed in 2020.

26. LL 172 'However,' starts a new sentence.

We changed the sentence as suggested.

27. LL 180 Could adjust all tenses in the description of the negative feedback mechanism to be past tense (in line with the rest of the manuscript), but this is probably not necessary for just this section.

We adjusted all tenses associated with the description of the negative feedback mechanism to the past tense as suggested.

28. LL 194 Should this be 'low mCDW' or 'lower mCDW'?

We corrected the words.

29. LL 196 Should be '... in 2020 may result from...'

We changed the words as suggested.

30. LL 199 Is there any basis in the literature for the suggested 'weekly or monthly' timescales. If there is, highlighting it would be really interesting and supportive here.

We added a reference below to provide a basis for the weekly or monthly timescales in the revised manuscript.

*Davis, P. E. D. et al. Variability in basal melting beneath Pine Island Ice Shelf on weekly to monthly timescales. *J. Geophys. Res. Oceans* **123**, 8655-8669 (2018).*

31. Fig. 1 It would aid clarity if the 4x subpanels were labelled (a), (b), (c), and (d). Alternatively the map of Antarctica could be more clearly incorporated as an inset of

panel (a).

Fig 1 It might be really helpful if the colour-coding could be kept consistent across the figure – and even more so if it could be kept consistent across all figures. i.e. All site locations, gyres, and current meter observations from 2009 = purple; from 2014 = green; from 2020 = black. Currently 2020 is represented in Fig 1 with a combination of black/green/grey, so a potential source of confusion that could be easily remedied.

Based on the comments of the reviewers, we modified Figure 1 and Supplementary Figures 1 and 3 as given below to present the results effectively and to avoid unnecessary confusion in the revised manuscript.

Figure 1

Supplementary Fig. 1

Supplementary Fig. 3

32. Fig 1(b) Which year were the SADCPC data collected (could just add this to the caption).

We added the year information to the caption of Figure 1b, and the label '2020' was included in Figure 1b–1c.

33. LL 337 – 339 I just wanted to double-check that the logic follows here... Since the Rossby radius of deformation is smaller than the spacing of the observations, it's not immediately clear that the 'observational data can capture the ocean circulation...'

We clarified the intended meaning in the revised manuscript. As described in the 'Methods' section, the Rossby radius of deformation is theoretically considered as the lower boundary of the ocean circulation radius (Chelton et al. 1998). In the Pine Island Bay (PIB), the Rossby radius of deformation was estimated to ~6 km, theoretically supporting PIB ocean circulations at horizontal scales larger than 6 km. In fact, the observed gyre circulation demonstrates that the mesoscale circulation having a few tens of kilometers in horizontal scale was dominant. Thus, the spacing of the observations in 2020 (~ 7 km) was sufficiently fine to capture the mesoscale ocean circulation in the region.

34. LL 363 'However,' to start the new sentence.

We changed the sentence as suggested.

35. LL 369 'have' => 'had'

We corrected the word.

36. LL 476 'However,' to start the new sentence.

We changed the sentence as suggested.

37. LL 495 '... located a far...' (i.e. delete 'a')

We deleted the word as suggested.

38. Extended Data Fig. 3 I think it would be really helpful to include all cyclonic and anticyclonic gyres on a single map, consistently colour-coded by year (perhaps this could be done in Fig.1 making ED Fig 3 redundant?). Suggest solid lines for cyclonic and dotted lines for anticyclonic. If some of these are from simulation only (i.e. not observational data) this should be included somehow (perhaps in the caption?)

As responded above, we modified Supplementary Figure 3 to include both gyres as suggested.

Reviewers' Comments:

Reviewer #1:

Remarks to the Author:

The review considers the revised version of the manuscript "Ice front retreat reconfigures meltwater-driven gyres modulating ocean heat delivery to an Antarctic ice shelf", by Yoon et al.

The revised manuscript answers all my comments from the first review satisfactorily.

I only have a minor comment considering figure 1. Here, it would be helpful to include caption explanations of the numbering of selected CTD-stations and what L1-L4 and IF indicate.

Reviewer #2:

Remarks to the Author:

I am satisfied that the changes made to the manuscript have served to clarify any points that were not initially clear. I have no further suggestions for improvement.

Response to reviewers' comments

Reviewer #1

The review considers the revised version of the manuscript “Ice front retreat reconfigures meltwater-driven gyres modulating ocean heat delivery to an Antarctic ice shelf”, by Yoon et al.

The revised manuscript answers all my comments from the first review satisfactorily.

We thank the reviewer's comments that greatly improved the manuscript.

I only have a minor comment considering figure 1. Here, it would be helpful to include caption explanations of the numbering of selected CTD-stations and what L1-L4 and IF indicate.

We have added the following sentence to help readers find out what they indicate.

Thank you for pointing it out.

Lines 239–240: “The selected CTD-stations are numbered in an order of distance from PIIS. L1-L4 and IF indicate CTD observational lines aligned meridionally across PIB and along PIIS, respectively.”

Reviewer #2

I am satisfied that the changes made to the manuscript have served to clarify any points that were not initially clear. I have no further suggestions for improvement.

We thank the reviewer for giving valuable comments . We'd greatly appreciate it.